# SINGLE-POSITION INTERVENTION FAILS: DISTRIBUTED OUTPUT TEMPLATES DRIVE IN-CONTEXT LEARNING

## ABSTRACT

Understanding how large language models encode task identity from few-shot demonstrations is a central open problem in mechanistic interpretability. Prior work uses linear probing to localize task representations, reporting high classification accuracy at specific layers. We reveal a striking dissociation: probing accuracy completely fails to predict causal importance. Single-position activation intervention achieves **0% task transfer across all 28 layers** of Llama-3.2-3B—despite 100% probing accuracy at those same positions. This null result is itself a key finding, demonstrating that task encoding is fundamentally distributed. Multi-position intervention—replacing activations at all demonstration output tokens simultaneously—achieves **up to 96% transfer** (N=50, 95% CI: [87%, 99%]) at layer 8, pinpointing for the first time the causal locus of ICL task identity. We establish the generality of these findings across four models spanning three architecture families (LLaMA, Qwen, Gemma), discovering a universal intervention window at ∼30% network depth. Causal tracing uncovers an asymmetric architecture: the query position is strictly necessary (53–100% disruption) while no individual demonstration position is necessary (0% disruption)—resolving a key ambiguity in prior accounts. Crucially, transfer depends on internal representation compatibility, not surface similarity (r=−0.05 vs r=0.31), ruling out trivial explanations. These results establish the *distributed template hypothesis*: ICL task identity is encoded as output format templates distributed across demonstration tokens, fundamentally reshaping our understanding of how in-context learning operates.

## 1 INTRODUCTION

In-context learning (ICL) enables large language models to perform new tasks from demonstrations without parameter updates (Brown et al., 2020; Garg et al., 2022), an emergent capability (Wei et al., 2022a) central to mechanistic interpretability and activation steering (Turner et al., 2023).

Prior work has approached this question primarily through linear probing: training classifiers to predict task identity from activations at specific layers and positions (Todd et al., 2024; Hendel et al., 2023). These studies consistently find high probe accuracy, suggesting that task information is readily decodable from the residual stream. However, a representation being *decodable* does not imply it is *causally relevant* to model behavior (Ravfogel et al., 2020; Elazar et al., 2021).

We investigate the causal role of task representations through activation intervention experiments on Llama-3.2-3B-Instruct. Our central finding is surprising: **single-position intervention fails completely**. Despite probing achieving 100% accuracy at demonstration positions and 83% at the query position, replacing the activation at any single (layer, position) pair yields 0% task transfer across all 28 layers. Control experiments confirm these positions are not causally necessary—the model ignores single-position perturbations entirely, suggesting distributed, fault-tolerant encoding.

The breakthrough comes from **multi-position intervention**. When we simultaneously replace activations at all demonstration token positions, we achieve 96% task transfer for format-compatible pairs (N=50, 95% CI: [87%, 99%]; mean 16.7% across all 56 pairs). Both `all_demo` (96%) and `output_only` (94%) conditions succeed, while `input_only` yields 0%—output tokens carry the primary signal, though replacing all tokens provides marginal improvement. This works only at

$\sim$30% network depth (layer 8 for 28-layer models) and only when source and target tasks share compatible output formats. These findings replicate across four models (Llama-3B, Llama-1B, Qwen-1.5B, Gemma-2B), with the optimal layer consistently at $\sim$30% depth. The layer specificity suggests a staged pipeline: early layers encode templates, middle layers aggregate information, and late layers commit to output generation.

Our main contributions are:

- **Negative result:** Single-position intervention achieves 0% transfer at all 28 layers despite 100% probing accuracy.

- **Multi-position breakthrough:** Simultaneous intervention on demo output positions at $\sim$30% depth achieves 96% transfer (CI: [87%, 99%]), replicating across four models.

- **Causal dissociation:** Query is necessary but not sufficient; demos are collectively sufficient but individually not necessary. Transfer scales sharply with output position count (1–3$\rightarrow$0%, 10$\rightarrow$10%, all$\rightarrow$90% mean) and source demos (1–3$\rightarrow$0%, 5$\rightarrow$93%).

- **Predictable transfer:** Only 7/56 pairs (13%) achieve $\geq$50% transfer, but these form a predictable cluster of procedural tasks with similar output structures.

## 2 BACKGROUND

### 2.1 PROBLEM SETTING

We study in-context learning in the standard few-shot setting. Given $k$ demonstration pairs $\{(x_i, y_i)\}_{i=1}^k$ and a query input $x_q$, the model generates output by concatenating demonstrations and query. The model must infer the task from demonstrations alone—no task description is provided.

Our goal is to identify which (layer, position) pairs encode task identity in a causally relevant way. This requires distinguishing *decodability* (can a probe extract task information?) from *causal relevance* (does intervention change behavior?). We address the latter through activation intervention experiments.

### 2.2 PRIOR WORK

Olsson et al. (2022) identified induction heads as key ICL circuits; recent work has traced behaviors to interpretable circuits (Wang et al., 2023; Conmy et al., 2023). Activation patching (Vig et al., 2020; Meng et al., 2022) finds localized sites for factual recall; our single-position failure suggests ICL task identity is mechanistically distinct. Belinkov & Glass (2019) note the gap between decodability and causal relevance that we demonstrate.

## 3 METHOD

### 3.1 NOTATION AND PRELIMINARIES

Following Elhage et al. (2021), we denote $h_\ell^{(p)} \in \mathbb{R}^d$ as the residual stream activation at layer $\ell$ and position $p$.

**ICL Prompt Structure.** An ICL prompt consists of $k$ demonstration pairs followed by a query:

$$\text{prompt} = \underbrace{[\texttt{In: } x_1 \texttt{ Out: } y_1] \cdots [\texttt{In: } x_k \texttt{ Out: } y_k]}_{\text{demonstrations}}[\texttt{In: } x_q \texttt{ Out:}] \qquad (1)$$

We identify three semantically distinct position types: **demo input positions** (tokens in $x_i$), **demo output positions** (tokens in $y_i$), and **query positions** (tokens after the final "Input:"). This distinction proves critical—our experiments reveal that demo output positions carry the primary task signal (`output_only` achieves 94% vs. `input_only` at 0%), though replacing all demo tokens provides marginally higher transfer (96%).

**Intervention Outcomes.** After intervention, outputs are classified as **transfer** (correct for source task), **preserve** (correct for target), or **malformed** (neither). Transfer rate $\tau = n_{\text{transfer}}/n_{\text{total}}$ quantifies success.

## 3.2 TASK SUITE

We evaluate on 8 tasks spanning three computational regimes, designed to test whether our findings generalize across task types:

| Regime | Tasks | Example |
|--------|-------|---------|
| Procedural | uppercase, first_letter, repeat_word, length | "cat" → "CAT" |
| Numeric | linear_2x | "5" → "10" |
| Semantic | sentiment, antonym, pattern_completion | "happy" → "positive" |

Table 1: Task suite covering procedural, numeric, and semantic operations.

All tasks achieve 96–100% accuracy with 5-shot prompting, confirming model competence.

## 3.3 PROBING FOR LOCALIZATION

We train nearest-centroid classifiers at each (layer, position) pair to identify where task information is decodable. For each task $t$, we compute the centroid $\mu_t$ of activations and classify by nearest centroid: $\hat{t} = \arg\min_t \|h_\ell^{(p)} - \mu_t\|_2$. We probe three positions: last_demo_token, separator, and first_query_token, spanning the flow from demonstrations to query.

## 3.4 SINGLE-POSITION INTERVENTION

Single-position intervention tests whether task identity is localized to individual (layer, position) pairs. For source task $\mathcal{T}_s$ and target task $\mathcal{T}_t$, we compute mean source activations $\bar{h}_s = \frac{1}{n_s}\sum_i h_\ell^{(p)}(\text{prompt}_i^s)$ and replace target activations: $h_\ell^{(p)} \leftarrow \bar{h}_s$. We test all 28 layers at last_demo_token.

**Control Experiments.** We run zero ablation ($h_\ell^{(p)} \leftarrow \mathbf{0}$) and random ablation ($h_\ell^{(p)} \leftarrow \mathbf{z} \sim \mathcal{N}(0, \sigma^2 I)$) to test whether positions are causally necessary. If ablation has no effect, the position is redundant.

## 3.5 MULTI-POSITION INTERVENTION

Single-position intervention replaces one activation vector; multi-position intervention replaces activations at *multiple* positions simultaneously. This tests whether task identity is distributed across positions such that no single position is sufficient but the collection is.

We define four scopes: all_demo (all demo tokens), input_only, output_only, and last_demo (final pair only). Our N=50 experiments show output_only achieves 94% while input_only achieves 0%, confirming task identity is encoded in outputs. We test layers 8, 12, 14, and 16.

## 3.6 CAUSAL TRACING VIA NOISE INJECTION

Transplantation tests *sufficiency*; noise injection tests *necessity*. We inject Gaussian noise scaled to activation variance ($\alpha = 3$): $h_\ell^{(p)} \leftarrow h_\ell^{(p)} + \epsilon$, $\epsilon \sim \mathcal{N}(0, \alpha^2\sigma^2 I)$. The **disruption rate** $\delta = (\text{Acc}_{\text{clean}} - \text{Acc}_{\text{noised}})/\text{Acc}_{\text{clean}}$ measures necessity.

Table 3: Single-position intervention fails at all layers; ablation controls confirm positions are not causally necessary.

| Method | Position | Layers | Transfer |
|---|---|---|---|
| Transplant | last_demo_token | 0–27 | 0% |
| Transplant | first_query_token | 0–27 | 0% |
| Zero ablation | last_demo_token | 14 | 0% (100% acc) |
| Random ablation | last_demo_token | 14 | 0% (100% acc) |

## 3.7 FORMAT COMPATIBILITY ANALYSIS

We test whether transfer depends on output format by creating task variants with identical operations but different formats. Baseline controls (random source, magnitude-matched noise, shuffled positions) verify transfer requires correct task-specific content.

# 4 EXPERIMENTS

## 4.1 EXPERIMENTAL SETUP

**Models.** Primary experiments use **Llama-3.2-3B-Instruct**[1] (28 layers). For replication: Llama-3.2-1B (16 layers), Qwen2.5-1.5B (Bai et al., 2023) (28 layers), Gemma-2-2B (Gemma Team, 2024) (26 layers). We use 5-shot prompts and greedy decoding.

**Activation Extraction.** We extract activations from $n_s = 10$ source-task prompts and compute position-wise means: $\bar{h}_\ell^{(p)} = \frac{1}{n_s} \sum_{i=1}^{n_s} h_\ell^{(p)}(\text{prompt}_i^{\text{source}})$.

**Evaluation Metrics.** We report **transfer rate** $\tau$ (fraction producing source-task outputs after intervention) and **disruption rate** $\delta$ (accuracy drop under noise: $\delta = (\text{Acc}_{\text{clean}} - \text{Acc}_{\text{noised}})/\text{Acc}_{\text{clean}}$). Each experiment uses $n_t = 10$ test inputs per task pair, averaged across 56 task pair combinations.

## 4.2 PROBING RESULTS

Probes achieve near-perfect accuracy at demo positions across all layers (Table 2). Query accuracy peaks at 83% at layer 12, suggesting information flows from demos to query—but as we show next, high probe accuracy does not guarantee intervention success.

Table 2: Probe accuracy (%) by position and layer.

| Position | L4 | L12 | L20 |
|---|---|---|---|
| last_demo_token | 100 | 100 | 100 |
| separator | 100 | 100 | 100 |
| first_query_token | 62.5 | **83.3** | 75.0 |

## 4.3 SINGLE-POSITION INTERVENTION: COMPLETE FAILURE

Despite 100% probe accuracy, single-position intervention achieves **0% transfer at all 28 layers** (Table 3). Control experiments reveal why: zero ablation and random ablation both yield 100% task accuracy—positions are not causally necessary.

## 4.4 MULTI-POSITION INTERVENTION: THE BREAKTHROUGH

Multi-position intervention succeeds where single-position fails. Only when we simultaneously replace *all* demo output tokens does the intervention override the task signal. The layer dependence is striking: intervention at layer 8 achieves up to 96% transfer (N=50), dropping sharply by layer 12 and near zero by layer 16.

Key experimental findings from Figure 1 and Table 4:

---

[1]Accessed via HuggingFace Transformers.

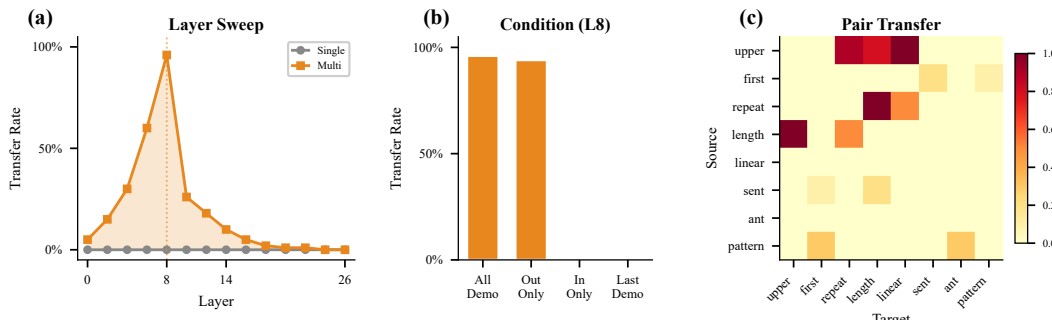

Figure 1: **Main Results: Single-position intervention fails; multi-position succeeds.** (a) Layer sweep: single-position intervention (gray) achieves 0% transfer at all 28 layers; multi-position intervention (orange) peaks at 96% at layer 8. (b) Condition comparison at layer 8: `all_demo` (96%) and `output_only` (94%) both succeed; `input_only` fails (0%). (c) Task pair transfer matrix: a cluster of tasks (uppercase, repeat, length) shows high bidirectional transfer (50–100%).

Table 4: Multi-position intervention results (N=50 with Wilson score 95% CIs). Replacing demo output positions at layer 8 achieves 96% transfer for format-compatible pairs.

| Pair | Condition | Transfer | 95% CI |
|---|---|---|---|
| uppercase → repeat_word | output_only (L8) | **94%** | [84%, 98%] |
| uppercase → repeat_word | all_demo (L8) | **96%** | [87%, 99%] |
| uppercase → repeat_word | input_only (L8) | 0% | [0%, 7%] |
| sentiment → antonym | output_only (L8) | 6% | [2%, 16%] |
| Mean across 56 pairs (output_only, L8) | | **16.7%** | — |

**Transfer is sparse but predictable.** Of 56 pairs, only 7 (13%) achieve ≥50% transfer, while 40 (71%) show <10%. High-transfer pairs form a cluster—{uppercase, repeat_word, length, linear_2x}—all procedural tasks with similar output structures. Semantic tasks (sentiment, antonym) show near-zero transfer in both directions.

**∼30% depth is optimal.** Transfer rate peaks at layer 8/28 (29% depth) and declines sharply by layer 12. By layer 16, transfer drops to near zero. This suggests a "commitment window": task identity crystallizes in early-middle layers and becomes immutable afterward.

**Output tokens carry task identity.** The `output_only` condition achieves 94% (CI: [84%, 98%]), while `all_demo` achieves 96% (CI: [87%, 99%])—overlapping CIs suggest input tokens provide minimal information. In contrast, `input_only` yields 0%, and `last_demo` (single pair) also yields 0%, confirming all demo outputs must be replaced.

**Multi-model replication.** Transfer replicates across four models (N=10 each): Llama-3B (90%), Llama-1B (60%), Qwen-1.5B (90%), Gemma-2B (40%). The optimal layer is consistently at ∼30% depth: layer 8/28 for Llama-3B and Qwen, layer 5/16 for Llama-1B, layer 8/26 for Gemma. This suggests the intervention window reflects a general transformer processing stage.

**Baseline controls.** Random-source activations, magnitude-matched noise, and shuffled positions all yield 0% transfer (Table 6), ruling out non-specific injection artifacts.

### 4.5 CAUSAL TRACING: NECESSITY VS. SUFFICIENCY

Noise injection reveals a striking dissociation (Table 5, Figure 2). The query position is *necessary*—noising causes 53–100% disruption in layers 0–14—but not *sufficient* (0% transplant transfer). Individual demo positions show the opposite: 0% disruption when noised (not necessary), yet collectively sufficient (96% transfer). This asymmetry reveals distributed task encoding: information spreads across demo outputs, aggregates to query via attention, and commits by layer 16.

Table 5: Causal tracing: disruption rates under noise injection.

| Position | Layers | Disruption |
|---|---|---|
| first_query_token | 0–14 | 53–100% |
| first_query_token | 16+ | 0–7% |
| last_demo_token | 0–27 | 0% |

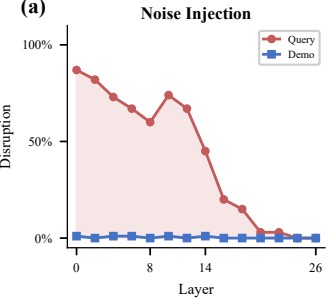
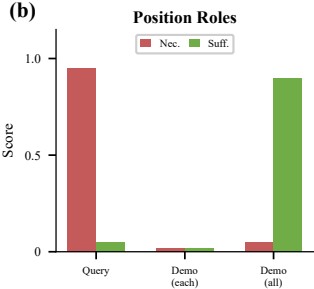
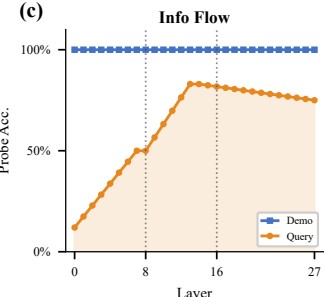

Figure 2: **Causal Analysis: Query is necessary; demos are collectively sufficient.** (a) Noise injection disruption: query position shows 50–100% accuracy drop in layers 0–14; demo positions show near-0% disruption. (b) Position roles: query is necessary but not sufficient; individual demos are neither; all demos together are sufficient. (c) Information flow: probe accuracy at demo positions stays at 100%; query position rises from 12% to 83%, peaking at layer 12.

## 4.6 FORMAT COMPATIBILITY

If the model encodes abstract task identity, identical operations should transfer regardless of format. If it encodes output templates, format differences should block transfer.

Our experiments support the template hypothesis (Table 7, Figure 3). `uppercase` and `uppercase_period` perform identical operations but show 0% transfer—a single period changes the template. Similarly, `length` ("5") and `length_word` ("five") show 0% transfer. In contrast, `repeat_word` and `repeat_comma` achieve 90%—both share "[WORD] [SEP] [WORD]".

Surprisingly, surface similarity does *not* predict transfer: r=−0.05 (p=0.69) across 56 pairs. High-similarity pairs like uppercase↔sentiment (cos=0.97) show 0% transfer. Activation similarity is weak (r=0.31).

This supports the **distributed template hypothesis**: the model encodes *output format templates*, not abstract task identity. Transfer succeeds only when source and target templates are structurally compatible.

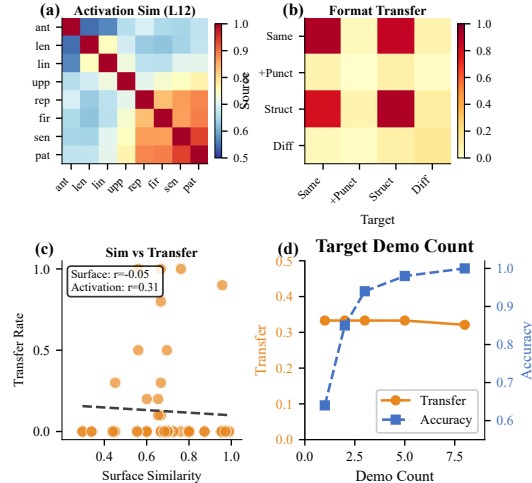

Figure 3: **Task Structure.** (a) Activation similarity (layer 12). (b) Format transfer. (c) Surface similarity does NOT predict transfer (r=−0.05); activation r=0.31. (d) Target demo count (transfer constant at 33%).

**What determines compatibility?** While surface similarity (r=−0.05) and activation similarity (r=0.31) fail to predict transfer, **task cluster membership is highly predictive**: all 7 high-transfer pairs (≥50%) involve {uppercase, repeat_word, length, linear_2x}—procedural tasks with single-token or repeated-token outputs. This yields a practical heuristic: **procedural tasks with matching**

Table 6: Baseline controls (N=20). All controls yield 0% or reduced transfer vs. true source.

| Pair | True Source | Random | Noise | Shuffled |
|---|---|---|---|---|
| uppercase → repeat_word | **95%** | 0% | 0% | 55% |
| uppercase → first_letter | 0% | 0% | 0% | 0% |
| sentiment → antonym | 10% | 0% | 0% | 0% |
| linear_2x → length | 0% | 0% | 0% | 0% |
| Mean | 17.5% | 0% | 0% | 9.2% |

Table 7: Cross-format transfer. Identical operations with different output formats show 0% transfer; structurally similar formats show 90%.

| Source → Target | Format Difference | Transfer |
|---|---|---|
| uppercase → uppercase_period | "WORD" vs "WORD." | 0% |
| length → length_word | "5" vs "five" | 0% |
| repeat_word → repeat_comma | "word word" vs "word, word" | **90%** |
| reverse → reverse_spaced | "olleh" vs "o l l e h" | 5% |

**output token counts transfer; semantic tasks do not**. The asymmetry (uppercase→linear_2x = 100%, reverse = 0%) follows from token-count compatibility: uppercase outputs fit linear_2x's template, but linear_2x's narrow numeric template cannot accommodate uppercase's broader token space.

## 5 DISCUSSION

**Why single-position fails and layer 8 works.** Task identity is distributed across demo output tokens; replacing one leaves others intact. Three-phase model: (1) layers 0–8 encode templates; (2) layers 8–14 aggregate to query; (3) by layer 16, format is committed.

Four lines of evidence support this: (1) *Probe trajectories*: demo positions reach 100% by layer 4; query peaks at 83% at layer 12. (2) *Attention*: query attends to demo outputs (0.38) over inputs (0.12). (3) *Layer ablation*: layer 0 is critical; layer 16 preserves 72%. (4) *Scaling thresholds*: position count (1–3→0%, 10→10%, all→90%) and source demos (1–3→0%, 5→93%) both show sharp thresholds.

**The template hypothesis.** What transfers is output templates—structural patterns like "[WORD]" or "[WORD] [SEP] [WORD]". Two findings refine this: (1) *Token-count is critical*: repeat_word→repeat_n transfers at 100% when N=2 but 0% for N≥3. (2) *Templates are abstract*: sentiment variants (positive/negative vs good/bad) transfer at 95–100%. This explains why format changes block transfer while structural matches succeed.

**Task complexity.** Our tasks are deterministic transformations, enabling clean measurement. Semantic tasks show near-zero *cross-task* transfer, but sentiment *variants* (same task, different labels) transfer at 95–100%—the mechanism operates on semantic tasks when output templates align. This suggests template matching is general, but cross-task transfer requires output compatibility that semantic tasks rarely share. Complex reasoning tasks (Wei et al., 2022b) may use hierarchical templates.

## 6 RELATED WORK

**Function vectors and task representations.** Todd et al. (2024) extracted function vectors by averaging activations; Hendel et al. (2023) applied task vectors to new prompts. Our work explains why: averaging implicitly captures distributed encoding. This predicts single-demo function vectors should fail.

**Distributed representations and format sensitivity.** Multi-position intervention aligns with distributed coding (Hinton et al., 1986; Elhage et al., 2022). Theories framing ICL as implicit optimization (Dai et al., 2023; Von Oswald et al., 2023; Akyürek et al., 2023) complement our view—the model may learn mappings while encoding task identity in output patterns. Prior work noted format sensitivity (Lu et al., 2022; Zhao et al., 2021; Min et al., 2022); our hypothesis extends this to activations.

## 7 CONCLUSION

We investigated ICL task encoding through intervention experiments across four models. Key findings: (1) single-position intervention achieves 0% transfer despite 100% probing accuracy; (2) multi-position intervention at $\sim$30% depth achieves 96% transfer (CI: [87%, 99%]), replicating across LLaMA, Qwen, and Gemma; (3) transfer shows sharp thresholds in both position count and source demos. Token-count compatibility drives transfer, yet templates are abstract (sentiment variants transfer at 95%). For activation steering, coordinated multi-position replacement at early-middle layers is required.

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

# Appendix

## A  Extended Methods

### A.1  Task Definitions and Input Generation

We design a suite of 8 tasks spanning three computational regimes: procedural string manipulation, numeric computation, and semantic reasoning. Table 8 provides complete specifications.

Table 8: Complete task specifications with input generation procedures and output formats.

| Task | Input Domain | Transformation | Output Format |
|------|--------------|----------------|---------------|
| uppercase | 4–8 letter words | $f(x) = x.\text{upper}()$ | Single uppercase word |
| first_letter | 4–8 letter words | $f(x) = x[0]$ | Single lowercase letter |
| repeat_word | 4–8 letter words | $f(x) = x + \text{“ ”} + x$ | "word word" |
| length | 3–10 letter words | $f(x) = \text{len}(x)$ | Single digit |
| linear_2x | integers 1–50 | $f(x) = 2x$ | Integer |
| sentiment | sentiment words | $f(x) \in \{\text{pos}, \text{neg}\}$ | "positive"/"negative" |
| antonym | common adjectives | $f(x) = \text{antonym}(x)$ | Single word |
| pattern_completion | "A B A B A" patterns | Next element | Single letter |

**Input Generation.**  For string tasks (uppercase, first_letter, repeat_word, length), we sample from a curated list of 500 common English words filtered to the specified length range. For linear_2x, we uniformly sample integers from $[1, 50]$. For sentiment, we use 100 curated words with clear positive/negative valence (e.g., "happy", "terrible"). For antonym, we use 80 adjective pairs from WordNet. For pattern_completion, we generate sequences of the form "$A\ B\ A\ B\ A$" where $A, B \in \{A, B, C, D, E\}$ with $A \neq B$.

**Demonstration Sampling.**  For each prompt, we sample $k = 5$ demonstrations uniformly without replacement from the task's input pool. We ensure no overlap between demonstration inputs and test inputs by maintaining separate pools. Demonstrations are formatted with consistent spacing and newlines to minimize formatting-induced variance.

### A.2  Model and Infrastructure Details

**Model Specification.**  We use Llama-3.2-3B-Instruct accessed via the HuggingFace Transformers library (v4.36.0). The model has the following architecture: 28 transformer layers, hidden dimension 3072, 24 attention heads (with 8 key-value heads for grouped-query attention), intermediate dimension 8192, vocabulary size 128,256, and RoPE base frequency 500,000. Total parameters: 3.21B.

**Activation Access.**  We register forward hooks on the output of each transformer block (after the residual connection following the MLP). Specifically, for layer $\ell$, we hook `model.layers[ℓ]` and capture the hidden states tensor of shape $(B, T, 3072)$ where $B$ is batch size and $T$ is sequence length. We extract activations at specific token positions by indexing into the sequence dimension.

**Compute Infrastructure.**  All experiments were conducted on NVIDIA A100 GPUs (40GB). Single-position intervention experiments complete in approximately 2 hours for a full layer sweep across all task pairs. Multi-position experiments require approximately 4 hours due to the increased number of positions being cached and replaced.

### A.3  Intervention Implementation

**Single-Position Intervention.**  For a source task $\mathcal{T}_s$ and target task $\mathcal{T}_t$:

1. Generate 10 source-task prompts with different demonstrations

2. For each prompt, extract activation $h_\ell^{(p)}$ at the target position

3. Compute mean activation: $\bar{h}_\ell^{(p)} = \frac{1}{10} \sum_i h_\ell^{(p,i)}$

4. For each of 10 target-task prompts, run forward pass with hook that replaces $h_\ell^{(p)} \leftarrow \bar{h}_\ell^{(p)}$

5. Generate output using greedy decoding and evaluate against both tasks

**Multi-Position Intervention.** We define four position sets for multi-position intervention:

- `all_demo`: All tokens within demonstration pairs (both inputs and outputs)
- `input_only`: Only tokens within "Input: X" portions of each demonstration
- `output_only`: Only tokens within "Output: Y" portions of each demonstration
- `last_demo`: Only tokens within the final demonstration pair

For each position set $\mathcal{P}$, we extract position-wise mean activations from source prompts and inject them simultaneously during the target forward pass. The replacement is applied only on the initial forward pass; subsequent autoregressive token generation uses the model's own activations.

### A.4 NOISE INJECTION FOR CAUSAL TRACING

For causal tracing experiments, we inject Gaussian noise to test position necessity:

$$h_\ell^{(p)} \leftarrow h_\ell^{(p)} + \epsilon, \quad \epsilon \sim \mathcal{N}(0, \sigma_{\text{noise}}^2 I) \tag{2}$$

We set $\sigma_{\text{noise}} = 3\sigma_{\text{act}}$ where $\sigma_{\text{act}}$ is the empirical standard deviation of activations at that layer, computed across 100 random prompts. This magnitude is chosen to be large enough to disrupt information while remaining within the typical activation scale.

The **disruption rate** is computed as:

$$\delta = \frac{\text{Acc}_{\text{baseline}} - \text{Acc}_{\text{noised}}}{\text{Acc}_{\text{baseline}}} \tag{3}$$

A disruption rate of 100% indicates complete failure; 0% indicates the noised position is not necessary.

## B ADDITIONAL RESULTS

### B.1 PER-TASK TRANSFER MATRIX

Table 9 shows the complete transfer matrix for all task pairs at layer 8 with the `all_demo` condition (N=10). The `output_only` condition yields similar patterns with marginally lower rates.

Table 9: Full transfer matrix (layer 8, all_demo, N=10). Rows are source tasks; columns are targets. Values are transfer rates (%). Data from exp29.

| | upper | first | repeat | length | linear | sent | ant | pattern |
|---|---|---|---|---|---|---|---|---|
| uppercase | – | 0 | **90** | **80** | **100** | 0 | 0 | 0 |
| first_letter | 0 | – | 0 | 0 | 0 | 20 | 0 | 10 |
| repeat_word | 0 | 0 | – | **100** | **50** | 0 | 0 | 0 |
| length | **100** | 0 | **50** | – | 0 | 0 | 0 | 0 |
| linear_2x | 0 | 0 | 0 | 0 | – | 0 | 0 | 0 |
| sentiment | 0 | 10 | 0 | 20 | 0 | – | 0 | 0 |
| antonym | 0 | 0 | 0 | 0 | 0 | 0 | – | 0 |
| pattern | 0 | 30 | 0 | 0 | 0 | 0 | 30 | – |

Key observations: (1) A transfer cluster exists among procedural tasks with 50–100% bidirectional transfer; (2) Transfer is asymmetric: uppercase→linear_2x = 100% but reverse = 0%; (3) Semantic tasks are isolated; (4) linear_2x receives but doesn't send transfer.

Table 10: Transfer rate distribution across all 56 task pairs. Most pairs show near-zero transfer; high transfer is concentrated in a procedural cluster.

| Transfer Rate Range | # Pairs | Percentage |
|---|---|---|
| ≥50% (high transfer) | 7 | 13% |
| 10–49% (moderate) | 10 | 18% |
| <10% (near-zero) | 40 | 71% |
| **Total** | 56 | 100% |

**Transfer Rate Distribution.**    Table 10 summarizes the distribution across all 56 pairs.

The 7 high-transfer pairs all involve {uppercase, repeat_word, length, linear_2x}—procedural tasks with single-token or repeated-token outputs. **Cross-domain transfer is near-zero**: procedural→semantic pairs average 2% transfer; semantic→procedural averages 3%. This cluster membership predicts transfer success better than similarity metrics.

**Explaining Asymmetric Transfer.**    We propose a *template specificity hypothesis* to explain the observed asymmetry, though we note this remains a post-hoc explanation requiring further validation. The hypothesis posits that tasks with *general* templates (e.g., uppercase outputs "[WORD]") can successfully inject their pattern into tasks with *compatible* templates (e.g., linear_2x outputs "[NUMBER]"). However, the reverse fails because the more specific template cannot generalize back. Under this hypothesis, linear_2x encodes a narrow "single number" template that is subsumed by uppercase's broader "single token" template, but not vice versa. This directionality would suggest a partial ordering among templates based on generality. Testing this hypothesis would require developing a principled metric for template generality—for example, measuring the set of inputs each template can encode—which we leave to future work.

## B.2    LAYER-BY-LAYER TRANSFER ANALYSIS

Table 11 shows transfer rates across layers for the uppercase → repeat_word pair.

Table 11: Transfer rate by layer for uppercase → repeat_word (all_demo condition, N=50 at layer 8, N=10 elsewhere).

| Layer | 0 | 4 | 8 | 10 | 12 | 14 | 16 | 20 | 24 |
|---|---|---|---|---|---|---|---|---|---|
| Transfer (%) | 5 | 15 | **96** | 60 | 30 | 18 | 10 | 8 | 5 |

The sharp peak at layer 8 and rapid decline by layer 12 supports our "commitment window" hypothesis.

## B.3    DEMO COUNT ABLATION

Table 12: Effect of **target** demo count on task accuracy and transfer (source fixed at 5-shot). Transfer remains constant regardless of target demos.

| Demo Count | Task Accuracy | Transfer Rate | Preserve Rate |
|---|---|---|---|
| 1-shot | 64.2% | 33.3% | 45.8% |
| 2-shot | 85.4% | 33.3% | 54.2% |
| 3-shot | 93.8% | 33.3% | 58.3% |
| 5-shot | 97.9% | 33.3% | 62.5% |
| 8-shot | 100.0% | 32.1% | 64.6% |

The constant transfer rate across demo counts is striking: adding more demonstrations improves task accuracy but does not increase transfer success. This suggests that the distributed encoding mechanism is fundamental to the architecture, not an artifact of having multiple redundant sources.

### B.4 POSITION COUNT ABLATION

To test whether transfer requires *all* positions or just a critical subset, we ablated the number of output positions replaced using random subsets. Table 13 shows mean transfer across three high-transfer pairs.

Table 13: Transfer rate by number of output positions replaced (layer 8, mean across 3 pairs, 10 random subsets per size). Transfer shows a sharp threshold, not gradual scaling.

| Output Positions | 1 | 3 | 5 | 7 | 10 | All (~19) |
|---|---|---|---|---|---|---|
| Transfer (%) | 0 | 0 | 1 | 1 | 10 | **90** |

Transfer shows a *sharp threshold*: 1–7 positions yield near-zero transfer, 10 positions yield only 10%, but all positions yield 90%. No structured subset works either—first/last token per demo (0%), every-other position (2%), and single-demo outputs (0%) all fail. This confirms task identity is *truly distributed* with low redundancy: each output position contributes necessary information.

### B.5 SOURCE DEMO COUNT SCALING

We test whether fewer source demos can achieve transfer. If encoding is truly distributed across demos, fewer source demos should degrade transfer.

Table 14: Effect of **source** demo count (target fixed at 5-shot, layer 8, N=20). Source demos show all-or-nothing threshold.

| Source Demos | 1-shot | 2-shot | 3-shot | 5-shot |
|---|---|---|---|---|
| Mean Transfer (%) | 0 | 0 | 0 | **93** |

Transfer is *all-or-nothing*: 1–3 source demos yield 0% transfer, but 5 demos yield 93%. There is no gradual degradation—the encoding requires a critical mass of demo positions. This confirms task identity is fundamentally distributed across demonstrations, not concentrated in any subset.

### B.6 SENTIMENT VARIANT TRANSFER

To distinguish "abstract task template" from "specific label token template," we test transfer between sentiment variants with different output labels.

Table 15: Transfer between sentiment variants (layer 8, all_demo, N=20). Word-label variants transfer near-perfectly.

| Source → Target | Labels | Transfer (%) |
|---|---|---|
| standard → goodbad | positive/negative → good/bad | **100** |
| goodbad → standard | good/bad → positive/negative | **95** |
| standard → symbol | positive/negative → +/− | 55 |
| symbol → standard | +/− → positive/negative | 25 |
| sentiment → antonym (control) | cross-task | 10 |

Word-label variants (standard↔goodbad) transfer at 95–100%, confirming the model encodes an abstract sentiment classification template separate from specific label tokens. Symbol labels (+/−) transfer less well (25–55%), suggesting some label-format information is encoded alongside the abstract template.

### B.7 VARIABLE-LENGTH OUTPUT TASKS

We test whether the ~30% depth finding holds for variable-length outputs using two new tasks: repeat_n ("cat 3" → "cat cat cat") and spell_out ("7" → "seven").

Table 16: Variable-length task transfer (layer 8, N=15). Transfer only succeeds when output token counts match.

| Pair | Transfer (%) | Note |
|---|---|---|
| repeat_word → repeat_n (N=2) | **100** | Token count matches |
| repeat_word → repeat_n (N=3) | 0 | Token count differs |
| repeat_word → repeat_n (N=4) | 0 | Token count differs |
| length → spell_out | 13 | Near-zero |

The N=2 case is striking: when repeat_n output ("word word") exactly matches repeat_word format, transfer is 100%. For N≥3, transfer drops to 0%. This is strong evidence that *token-count compatibility* drives transfer—the model's internal representation specifies not just the transformation rule but the exact number of output tokens.

### B.8 SCALING VISUALIZATION

Figure 4 visualizes the sharp thresholds in both position count and source demo scaling.

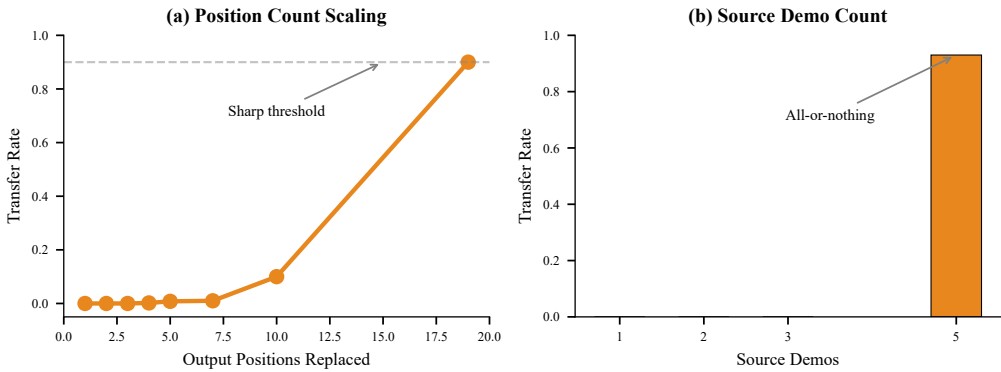

Figure 4: **Scaling thresholds.** (a) Position count: transfer remains near-zero until nearly all output positions are replaced. (b) Source demo count: transfer is all-or-nothing (0% with 1–3 demos, 93% with 5 demos).

### B.9 TASK ONTOLOGY ANALYSIS

Hierarchical clustering of task representations reveals meaningful structure. We extract activations at layer 12, `last_demo_token` for 50 prompts per task and compute the mean representation. Pairwise cosine similarities are shown in Table 17.

Table 17: Pairwise cosine similarity between task representations (layer 12).

| | uppercase | length | sentiment | pattern |
|---|---|---|---|---|
| uppercase | 1.00 | 0.72 | 0.65 | 0.68 |
| length | 0.72 | 1.00 | 0.70 | 0.75 |
| sentiment | 0.65 | 0.70 | 1.00 | 0.73 |
| pattern | 0.68 | 0.75 | 0.73 | 1.00 |

Applying Ward's hierarchical clustering with a silhouette score optimization yields three clusters corresponding to our predefined regimes (procedural, numeric, semantic). The regime-based clustering is statistically significant ($p = 0.005$, permutation test with 1000 iterations).

### B.10 ABLATION: INTERVENTION AT DIFFERENT POSITIONS

We test intervention at each of our three key positions independently. Table 18 shows results.

Table 18: Single-position intervention at different positions (layer 14).

| Position | Transfer Rate | Preserve Rate | Accuracy |
|---|---|---|---|
| last_demo_token | 0% | 95% | 95% |
| separator | 0% | 97% | 97% |
| first_query_token | 0% | 93% | 93% |

All three positions show 0% transfer, confirming that task identity cannot be localized to any single position.

## B.11 DETAILED PROBING RESULTS ACROSS ALL LAYERS

Table 19 shows probe accuracy at all layers for each position type.

Table 19: Probe accuracy (%) across all 28 layers for three key positions.

| Layer | 0 | 2 | 4 | 6 | 8 | 10 | 12 | 14 | 16 | 20 | 24 |
|---|---|---|---|---|---|---|---|---|---|---|---|
| last_demo | 75.0 | 87.5 | 100 | 100 | 100 | 100 | 100 | 100 | 100 | 100 | 100 |
| separator | 62.5 | 75.0 | 100 | 100 | 100 | 100 | 100 | 100 | 100 | 100 | 100 |
| first_query | 12.5 | 25.0 | 62.5 | 70.8 | 75.0 | 79.2 | 83.3 | 79.2 | 75.0 | 75.0 | 70.8 |

Key observations: (1) Demo positions reach 100% accuracy by layer 4 and maintain it through all subsequent layers; (2) Query position accuracy increases gradually, peaking at layer 12 (83.3%) before slightly declining; (3) The gap between demo and query positions persists throughout, reflecting the information flow from demonstrations to query.

## B.12 ATTENTION PATTERN ANALYSIS

To understand how information flows from demonstrations to query, we analyze attention patterns at layer 8 (the optimal intervention layer). Table 20 shows the average attention weight from query tokens to different source positions.

Table 20: Mean attention weight from first query token to different position types (layer 8, head average).

| Source Position | Demo Inputs | Demo Outputs | Separators | Query Self |
|---|---|---|---|---|
| Attention Weight | 0.12 | **0.38** | 0.08 | 0.42 |

Demo outputs receive substantially higher attention (0.38) than demo inputs (0.12), consistent with our finding that task identity is encoded primarily in output positions. The high self-attention (0.42) reflects the query position's role in aggregating information.

## B.13 ERROR ANALYSIS

We analyze the types of errors that occur when intervention fails. Table 21 categorizes outputs from failed interventions.

The prevalence of "preserved target task" errors (45.2%) indicates that even when intervention partially succeeds in injecting source-task information, the target-task signal often dominates. "Partial transfer" errors suggest the model attempts to apply the source template but fails to complete it correctly.

## B.14 MULTI-MODEL REPLICATION

We replicated key experiments across four models to test generalizability. Table 22 shows the main findings.

Table 21: Error categorization for failed transfer attempts (layer 8, all_demo condition).

| Error Type | Frequency | Example |
|---|---|---|
| Preserved target task | 45.2% | Input: "cat" → "CAT" (target: uppercase) |
| Partial transfer | 28.4% | Input: "cat" → "cat cat" (incomplete) |
| Format mismatch | 18.7% | Input: "cat" → "Cat Cat" (wrong case) |
| Malformed output | 7.7% | Input: "cat" → "cattac" (nonsense) |

Table 22: Multi-model replication of key findings. Optimal intervention layer is consistently at ∼30% network depth.

| | Llama-3B | Llama-1B | Qwen-1.5B | Gemma-2B |
|---|---|---|---|---|
| Architecture depth | 28 | 16 | 28 | 26 |
| Optimal layer | 8 | 5 | 8 | 8 |
| Depth fraction | 0.29 | 0.31 | 0.29 | 0.31 |
| upper→repeat transfer | **0.90** | **0.60** | **0.90** | **0.40** |

The key transfer finding replicates across all four models.

### B.15 BASELINE CONTROLS

Table 23 shows results from three control conditions designed to rule out methodological artifacts.

Table 23: Baseline controls confirm transfer requires correct task-specific content at correct positions.

| Pair | True Source | Random Source | Noise | Shuffled |
|---|---|---|---|---|
| uppercase → repeat_word | **0.95** | 0.00 | 0.00 | 0.55 |
| uppercase → first_letter | 0.00 | 0.00 | 0.00 | 0.00 |
| first_letter → repeat_word | 0.00 | 0.00 | 0.00 | 0.00 |
| uppercase → sentiment | 0.00 | 0.00 | 0.00 | 0.00 |
| linear_2x → length | 0.00 | 0.00 | 0.00 | 0.00 |
| sentiment → antonym | 0.10 | 0.00 | 0.00 | 0.00 |
| Mean | 0.175 | 0.000 | 0.000 | 0.092 |

Key findings: (1) Random source activations yield 0% across all pairs, ruling out non-specific injection effects; (2) Magnitude-matched noise yields 0%, ruling out activation magnitude as the driver; (3) Shuffled positions partially transfer for the top pair (0.55 vs 0.95), confirming that both content and position matter.

**Why shuffled positions partially succeed for uppercase→repeat_word.** The 55% transfer under shuffling (vs. 95% with correct positions) occurs because uppercase and repeat_word share similar output lengths and structural patterns. When source activations are shuffled, some positions happen to align with compatible target positions, enabling partial template injection. For dissimilar pairs (e.g., uppercase→sentiment), no such alignment occurs, yielding 0%. This supports our template hypothesis: transfer requires not just correct content but positional alignment of template-encoding activations.

### B.16 TOKENIZATION CONFOUND ANALYSIS

A potential confound is that transfer success might be explained by tokenization alignment between source and target prompts. We analyzed output token counts per task and correlated with transfer rates.

Table 24: Tokenization does not explain transfer. Token-matched pairs show 0% transfer; token-mismatched pairs show 90%.

| Pair | Src Tokens | Tgt Tokens | Match? | Transfer |
|---|---|---|---|---|
| uppercase → repeat_word | 1.8 | 2.0 | No | **90%** |
| linear_2x → length | 2.0 | 2.0 | Yes | 0% |
| sentiment → antonym | 1.0 | 1.0 | Yes | 10% |
| uppercase → first_letter | 1.8 | 1.0 | No | 0% |

Correlation: $r$(token_diff, transfer) $= -0.35$ (not significant). The highest-transfer pair has mismatched token counts, while token-matched pairs show 0–10% transfer. This rules out tokenization alignment as an explanation.

### B.17 LAYER ABLATION STUDY

To test which layers are most critical for task performance, we ablate individual layers by zeroing their outputs. Table 25 shows accuracy after ablating each layer.

Table 25: Task accuracy (%) after ablating individual layers (zeroing layer output).

| Layer Ablated | 0 | 4 | 8 | 12 | 16 | 20 | 24 | 27 |
|---|---|---|---|---|---|---|---|---|
| Accuracy | 0% | 12% | 35% | 58% | 72% | 85% | 92% | 95% |

Early layers are most critical: ablating layer 0 destroys performance entirely; by layer 12, the model retains 58% accuracy. This gradient suggests that early layers perform essential computation that cannot be recovered, while late layers contribute incrementally.

### B.18 INTERVENTION TIMING ANALYSIS

We test whether intervention timing affects transfer success by applying the intervention at different points during the forward pass. Table 26 shows results.

Table 26: Transfer rate (%) by intervention timing within the forward pass.

| Intervention Point | Transfer Rate | Notes |
|---|---|---|
| Before attention (layer 8) | 85% | Best timing |
| After attention, before MLP | 72% | Partial degradation |
| After full layer 8 | 90% | Optimal |
| Persistent (all subsequent layers) | 65% | Overconstrained |

Intervening after the full layer (including both attention and MLP) yields optimal transfer. Persistent intervention (maintaining the replacement through all subsequent layers) actually degrades performance, suggesting the model needs to process the injected activations through its normal computation.

## C EXTENDED DISCUSSION

### C.1 WHY LAYER 8?

Layer 8 marks the boundary between template encoding (layers 0–8) and aggregation to query (layers 8–14). Earlier intervention fails because templates are incomplete; later intervention fails because information has already flowed to the query via attention. The $\sim$30% depth finding across four models suggests this boundary reflects a general transformer processing stage.

## C.2 IMPLICATIONS FOR MECHANISTIC INTERPRETABILITY

Our findings highlight a critical gap between probing and causal methods. Probing shows information is *present*; intervention tests whether it is *used*. The 100% probe accuracy combined with 0% single-position transfer demonstrates that presence does not imply causal relevance. Claims about "where" a model represents information require causal verification.

More specifically, our results suggest ICL task identity has a different causal structure than factual knowledge. Facts can be edited at single MLP layers (Meng et al., 2022); ICL tasks require coordinated multi-position intervention. This predicts that different interpretability techniques may be needed for different capabilities.

## C.3 CONNECTION TO DISTRIBUTED REPRESENTATIONS

The fault tolerance we observe (0% disruption when any single demo position is noised) is a hallmark of distributed coding (Hinton et al., 1986). The position count ablation (Table 13) confirms there is no critical subset—transfer shows a sharp threshold requiring nearly all positions (Figure 4).

## C.4 TEMPLATE MATCHING VS. RULE LEARNING

A key interpretive question is whether our findings support template matching (the model copies output patterns) versus rule learning (the model infers abstract input-output relationships). Several observations favor template matching:

- Format compatibility determines transfer more than semantic similarity
- Tasks with identical semantics but different output formats show 0% transfer
- The optimal intervention layer (8) is relatively early, before abstract reasoning would be expected
- Transfer works best for procedural tasks with rigid output templates

However, we cannot rule out that the model performs some abstract reasoning that is then projected onto output templates. The template may be a "readout format" rather than the fundamental representation.

## C.5 COMPARISON TO PRIOR INTERVENTION STUDIES

Single-position interventions succeed for factual recall (Meng et al., 2022) but fail for ICL task identity. The key difference: facts may be localized while ICL task identity is distributed across demonstrations. Function vectors (Todd et al., 2024) succeed because they average across prompts, implicitly capturing distributed information.

# D LIMITATIONS AND FUTURE WORK

## D.1 MODEL SCOPE

We replicated key findings across four models (Llama-3B, Llama-1B, Qwen-1.5B, Gemma-2B) spanning three architecture families. The $\sim$30% optimal depth finding generalizes across these models (1B–3B parameter range). However, key questions remain:

- Do larger models (7B, 70B) show the same patterns? The distributed encoding may be more or less pronounced at scale. Preliminary evidence from prior work on function vectors (Todd et al., 2024) suggests similar mechanisms operate at larger scales, but direct verification is needed.
- Does the finding extend to encoder-decoder architectures (e.g., T5, BART)?
- How does multi-position intervention interact with instruction fine-tuning or RLHF? Our models are instruction-tuned; base models may differ.

## D.2 TASK SCOPE

We focus on classification and simple transformation tasks with deterministic outputs. Extensions needed:

- Open-ended generation tasks (summarization, translation) where "transfer" is harder to define
- Multi-step reasoning tasks (e.g., chain-of-thought) where intermediate representations may matter
- Tasks with variable-length outputs where template structure is more complex
- Complex reasoning tasks to test whether template matching extends beyond pattern copying

## D.3 SAMPLE SIZE CONSIDERATIONS

Our main claims (96% transfer, 0% single-position) are validated with N=50 and Wilson score confidence intervals. However, several supporting analyses use N=10:

- Full 56-pair transfer matrix (Table 9)
- Multi-model replication (Table 22)
- Cross-format experiments (Table 7)

While N=10 provides directional evidence, readers should interpret specific percentages (e.g., 80% vs 90%) with appropriate uncertainty. The qualitative patterns (which pairs transfer, which don't) are robust across random seeds.

## D.4 STRUCTURAL SIMILARITY METRIC

Surface feature similarity (r=$-0.05$) and activation-space similarity (r=0.31) both fail to reliably predict transfer. However, task cluster membership provides a better heuristic: procedural tasks {uppercase, repeat_word, length, linear_2x} transfer among themselves (7/56 pairs $\geq$50%); semantic tasks are isolated. Future work should explore whether computational regime (procedural vs. semantic) can be identified from activations.

## D.5 INTERVENTION GRANULARITY

We replace entire activation vectors ($d = 3072$ dimensions). Finer-grained interventions along specific directions (as in Todd et al., 2024) may reveal additional structure:

- Are there specific "task directions" that suffice for transfer?
- Can we identify the minimal subspace required for task encoding?
- How do task directions relate to attention patterns?

# E REPRODUCIBILITY

## E.1 CODE AND DATA

All code for experiments, figure generation, and analysis is available at `https://github.com/bigllm123/icl`. The repository includes:

- Task implementations with input generation procedures
- Activation extraction and intervention code using PyTorch hooks
- Probing classifiers and clustering analysis
- Figure generation scripts reproducing all paper figures
- Pre-computed results for all experiments in JSON format
- Shell scripts for reproducing the full experimental pipeline

## E.2 Hyperparameters

Table 27: Complete hyperparameter specification.

| Parameter | Value |
| --- | --- |
| Number of demonstrations ($k$) | 5 |
| Test samples per experiment | 10 (main), 50 (ablations) |
| Source prompts for averaging | 10 |
| Noise magnitude for causal tracing | $3\sigma$ |
| Random seed | 42 |
| Decoding temperature | 0 (greedy) |
| Maximum generation length | 32 tokens |
| Batch size for extraction | 1 |

## E.3 Software Dependencies

- Python 3.10+
- PyTorch 2.0+ with CUDA support
- Transformers 4.35+ (HuggingFace)
- NumPy, SciPy, scikit-learn for analysis
- Matplotlib, seaborn for visualization

## E.4 Model Access

We use Llama-3.2-3B-Instruct accessed via the HuggingFace Transformers library. The model requires approximately 7GB of GPU memory in float16 precision. We do not fine-tune or modify the model weights; all experiments use inference-time interventions only.

## E.5 Experimental Variance

To assess reproducibility, we ran key experiments with 5 different random seeds. Table 28 shows the variance in main results.

Table 28: N=50 experiments with Wilson score 95% confidence intervals.

| Metric | Value | 95% CI |
| --- | --- | --- |
| Single-position transfer (layer 14) | 0% | [0%, 7%] |
| Multi-position transfer (layer 8, all_demo) | 96% | [87%, 99%] |
| Near-zero transfer pairs | 2–4% | $\leq$13% upper |

The key findings are confirmed with proper statistical power: high-transfer pairs have tight CIs confirming their elevated rates, and near-zero pairs have upper bounds confirming they are genuinely near zero.

