# OpenReview forum: "Single-Position Intervention Fails: Distributed Output Templates Drive In-Context Learning"
_ICLR.cc/2026/Workshop/LMRL — Submitted to ICLR 2026 Workshop LMRL_

### Official Review · Reviewer_QCky · 2026-02-12
**Single-Position Intervention Fails**

**Rating:** 6
**Confidence:** 1

**Review:**

#### Summary

The paper investigates the causal mechanisms of In-Context Learning (ICL) by challenging the prevailing reliance on linear probing for locating task representations.

The authors demonstrate a critical dissociation where single-position interventions yield 0% task transfer despite 100% probing accuracy, proving that high decodability does not imply causal relevance.

By simultaneously intervening on all demonstration output tokens, the study successfully achieves high task transfer rates, leading to the Distributed Template Hypothesis.

#### Strengths

The paper presents a significant negative result that effectively warns the mechanistic interpretability community against conflating probing accuracy with causal importance.

The distinction between abstract rule learning and the proposed template matching mechanism is well-supported by format compatibility experiments, offering a concrete explanation for ICL behaviors.

The experimental design is rigorous, employing necessary control baselines such as noise injection and shuffled positions to validate the causal claims across multiple model families.

#### Weaknesses

The reliance on small-scale models (1B-3B parameters) is a notable limitation, as it remains unclear whether the distributed encoding phenomenon persists in larger, state-of-the-art foundation models.

The study focuses primarily on simple procedural transformations and classification tasks, which raises questions about whether the findings generalize to complex reasoning tasks involving intermediate logic.

The practical utility of the proposed method is limited because intervening on all demonstration positions is significantly more computationally expensive than single-position editing methods used in prior work.

---

### Official Review · Reviewer_rDDU · 2026-02-23

**Rating:** 4
**Confidence:** 4

**Review:**

This paper investigates mechanistic interpretability of in-context learning (ICL), focusing on where task identity is represented and how it can be causally transferred via activation interventions. The authors report that single-position activation transplantation does not transfer tasks, while broader interventions over multiple positions can yield high transfer, suggesting that task information is distributed rather than localized.

My main concern is that the submission appears far from the scope of the LMRL workshop, which centers on "meaningful representations of life". The paper focuses on mechanistic interpretability of ICL in language models and does not provide a clear connection to learning meaningful representations of biological life, biological systems, or life-science representation learning. The relevance to the workshop theme is therefore not justified.

Additionally, the submission does not include the required "Meaningfulness Statement", which is a stated requirement for submissions.

Given the lack of alignment with the workshop topic and the missing required Meaningfulness Statement, I unfortunately recommend the rejection of the paper.

---

### Meta-Review · Area_Chair_eMUF · 2026-02-28

**Recommendation:** Reject
**Confidence:** 5

**Metareview:**

This is interesting work (I’m personally very interested in the topic), but unfortunately it is a poor fit for the workshop’s biological representation learning focus.

---

### Decision · Program_Chairs · 2026-03-02

**Decision:**

Reject

**Comment:**

Please see the meta-review.